# Cardiac Arrest during Interventional Radiology Procedures: A 7-Year Single-Center Retrospective Study

**DOI:** 10.3390/jcm11030511

**Published:** 2022-01-20

**Authors:** In Chul Nam, Esther Sangeun Lee, Ji Hoon Shin, Vincent Xinrui Li, Hee Ho Chu, Sung Eun Park, Jung Ho Won

**Affiliations:** 1Department of Radiology, Gyeongsang National University College of Medicine, Gyeongsang National University Changwon Hospital, Changwon 51472, Korea; sky_hall@naver.com (I.C.N.); uneyes@hanmail.net (S.E.P.); 2Harvard College, Harvard University, Cambridge, MA 02138, USA; sangeunlee@college.harvard.edu (E.S.L.); vincentli@college.harvard.edu (V.X.L.); 3Department of Radiology and Research Institute of Radiology, University of Ulsan College of Medicine, Asan Medical Center, Seoul 05505, Korea; chuzzang1224@gmail.com; 4Department of Radiology, Gyeongsang National University College of Medicine, Gyeongsang National University Hospital, Jinju 52727, Korea; circlehoya@naver.com

**Keywords:** radiology, interventional, heart arrest, risk assessment

## Abstract

An intervention radiology (IR) unit collected cardiac arrest data between January 2014 and July 2020. Of 344,600 procedures, there were 23 cardiac arrest patients (0.0067%). The patient data was compared to a representative sample (N = 400) of the IR unit to evaluate the incidence and factors associated with cardiac arrest during IR procedures. Age, procedure urgency, American Society of Anesthesiologists (ASA) physical status, procedure type, and underlying medical conditions were identified as valuable predictors of a patient’s susceptibility to cardiac arrest during an IR procedure. The proportion of pediatrics was higher for cardiac arrest patients, and most required immediate procedures. The distribution of high ASA physical status (III or greater) was skewed compared to that of the non-cardiac arrest patients. Vascular procedures were associated with higher risk than non-vascular procedures. The patients who underwent non-transarterial chemoembolization arterial procedures demonstrated relative risks of 4.4 and 11.7 for cardiac arrest compared to biliary procedures and percutaneous catheter drainage, respectively. In addition, the six patients (26.1%) who died before discharge all underwent vascular procedures. Relative to patients with acute kidney injury, patients with malignancy, hypertension, and diabetes mellitus demonstrated relative risks of 3.3, 3.4, and 4.8 for cardiac arrest, respectively.

## 1. Introduction

Cardiac arrest is a detrimental event associated with multiple organ injury and dysfunction, traumatic complications, and a higher risk of death [1,2,3]. Although cardiac arrest has lasting adverse effects on the patients who undergo interventional radiology (IR) treatments, not enough research has been done on cardiac arrest cases that occur in IR suites. However, analyzing IR-related cardiac arrest is necessary, as it can highlight additional IR-associated risk factors previously overlooked in the literature. This study aimed to identify and evaluate such risk factors by conducting a retrospective study of cardiac arrest events in a single IR department.

## 2. Materials and Methods

This retrospective study investigated the cardiac arrest cases of the IR unit between 1 January 2014, and 31 July 2020. All reviewed cardiac arrest cases occurred in the IR suite, and the affected patients received cardiopulmonary resuscitation (CPR) from medical professionals at the institution. The medical records of the cardiac arrest patients were anonymized using eight-digit codes and were reviewed for clinical diagnoses, IR procedural data, cardiac arrest events, and resuscitation outcomes.

Each cardiac arrest event was noted in detail, including the time of the cardiac arrest event relative to the beginning of the IR procedure, the patient’s cardiac rhythms during the event (e.g., pulseless electrical activity, ventricular fibrillation, asystole), and the details of the CPR procedure, including the presence of defibrillation or intubation. Based on medical records, several factors were analyzed retrospectively. After identifying the patients’ underlying health conditions from the medical history and clinical diagnosis, the patients’ statuses, at the time of IR treatment, were classified retrospectively using the American Society of Anesthesiologists (ASA) physical status classification [4]. The primary cause of each cardiac arrest event was also speculated according to details documented in the respective operative reports.

To obtain a more objective view of the risk factors identified among cardiac arrest patients, overall statistics that encapsulated all procedures performed in the IR unit during the same period were compiled. However, the unit lacked a central system that collects statistical data. The only way to access patient data was to read individual medical charts. Since it would be impractical to review all medical charts for the 344,600 patients, a representative sample was selected and analyzed instead. Random sampling, based on the date of the procedure, was used for sample selection. Although a sample size of at least 100 patients is usually sufficient to summarize categorical variables for certain types of medical research [5], the sample size was chosen using a formula for sample size calculation for medical studies to optimize the present study results [6]. The sample size calculated using the formula was 384 patients, which was rounded to the nearest hundred (N = 400).

Several statistical tests were conducted using the cardiac arrest data and representative sample data from the IR unit. The two groups’ demographic data and ASA physical status data were compared using z-tests and chi-square tests of independence. The relative risks of cardiac arrest, based on underlying diseases, procedure types, and ASA physical status, were calculated by comparing the cardiac arrest incidence in each group. The procedures were further broken down into subcategories, and their relative risks were calculated. The significance of the relative risks was determined using two-proportion z-tests, and a chi-square test was conducted to evaluate the effect of vascular procedures on cardiac arrest incidence. Additionally, the cardiac arrest frequency in the IR unit was compared with the intraoperative cardiac arrest data reported by the American College of Surgeons National Surgical Quality Improvement Program (ACS-NSQIP) to evaluate the relevance of the characteristics of cardiac arrest cases that occurred in this IR unit.

The institutional board granted an ethical approval to conduct this study. The written informed consent requirement was waived due to the study’s retrospective design.

## 3. Results

During the study period, 344,600 procedures were conducted in the IR unit, and of those cases, 23 patients experienced cardiac arrest, yielding an incidence of 0.0067%. The demographic data of the cardiac arrest patients and the representative IR unit patients are presented in Table 1. There were 8.7% and 4.3% of the cardiac arrest patients from the intensive care unit and emergency department, respectively, and the remaining 20 patients (87%) were from the general ward.

There were no significant differences in gender and use of contrast medium during the IR procedures between the cardiac arrest and IR unit patient data. Although the mean ages of the cardiac arrest and IR unit sample patients were not statistically different, the proportions of patients in different age groups (infant, child, and adult) differed significantly between the cardiac arrest and IR unit patient data (*χ*^2^ = 24.24, df = 2, *p* < 0.001). The proportion of infants and children was significantly higher in the cardiac arrest patient data compared with the IR unit patient data. The frequency of urgent procedures was significantly higher in the cardiac arrest patient data than the IR unit patient data (17.4% vs. 1.5%) (*p* < 0.001).

All 23 patients were documented as class III or greater when classified according to the pre-procedural ASA physical status. Six patients (26.1%) were classified as class IV, and four (17.4%) as class V (Table 1). The mean ASA status of the patients who did not experience cardiac arrest was 2.8, while the mean ASA of the cardiac arrest patients was 3.6 (*p* < 0.001), and the chi-square test of independence revealed that the association between ASA status and cardiac arrest was statistically significant (*χ*^2^ = 104.47, df = 3, *p* < 0.001). Compared with ASA class III patients, class IV and V patients had relative risks of 9.1 and 22.4, respectively (*p* = 0.007, *p* = 0.001).

Table 2 summarizes our analysis of vascular vs. non-vascular cardiac arrest procedures. There were 24 procedures performed for 23 cardiac arrest patients; one patient underwent both a vascular and a non-vascular procedure. There were nineteen (79.2%) and five (20.8%) vascular and non-vascular procedures, respectively. Since the majority of patients who experienced cardiac arrest underwent vascular procedures, a chi-square test of independence was conducted to evaluate the association. The patient who underwent both a vascular and a non-vascular procedure was double counted for this test. The association between cardiac arrest and vascular procedures was statistically significant (*χ*^2^ = 5.09, df = 1, *p* = 0.024). All six patients who died before hospital discharge underwent vascular procedures. Non-transarterial chemoembolization (TACE) arterial procedures were associated with relative risks of 4.4 and 11.7 for cardiac arrest compared with biliary procedures and percutaneous catheter drainage, respectively (*p* = 0.0413, *p* = 0.025).

There were several commonly identified underlying diseases among the cardiac arrest patients. There were 17 patients (73.9%) diagnosed with malignancy, 13 (56.5%) with liver disease, and 9 (39.1%) with hypertension (HTN). Several patients were also diagnosed with chronic kidney disease, cardiovascular disease, and diabetes mellitus (DM) (21.7%, 21.7%, and 17.4%, respectively). Relative to other underlying diseases and conditions, acute kidney injury (AKI) was associated with a low risk of cardiac arrest; malignancy, HTN, and DM demonstrated relative risks of 3.3, 3.4, and 4.8 for cardiac arrest compared with AKI (*p* = 0.037, *p* = 0.0496, *p* = 0.030) (Figure 1).

Before their procedures, 3 patients (13.0%) experienced cardiac arrest, and 14 patients (60.9%) experienced cardiac arrest during their procedures. Shortly after their procedures, six patients (26.1%) experienced cardiac arrest. For seven patients (30.4%), cardiac arrest was attributed to allergic responses, four (17.4%) to hypovolemic shock, three (13.0%) to arrhythmia, two (8.7%) to oversedation, one (4.3%) to cardiac tamponade, one (4.3%) to an iatrogenic cause, and five (21.7%) to unknown causes. Of the 23 patients, 10 (43.5%) were defibrillated during CPR, and 15 (65.2%) were intubated.

While 17 cardiac arrest patients (73.9%) survived until hospital discharge, 6 cardiac arrest patients (26.1%) died before discharge. All of the patients who died in the hospital were intubated, but not all were defibrillated. On the same day as their procedures, five patients died, while one survived for one postoperative week. All six patients underwent vascular procedures: TACE (*n* = 1), non-TACE arterial procedures (*n* = 4), and venous central line insertion (*n* = 1). The median mean (±SD) age of the six patients who died was 65.5 (range, 38–79). None of the pediatric patients died.

## 4. Discussion

The IR unit’s cardiac arrest incidence in this study was 0.0067% (6.7 per 100,000 cases), which was 10 times lower than the intraoperative cardiac arrest incidence of 0.067% (6.7 per 10,000 cases) reported by the ACS-NSQIP data based on 1.3 million cases (*p* < 0.001) [7]. The IR unit’s cardiac arrest incidence of 0.0067% in this study was also significantly lower than another IR unit’s cardiac arrest incidence of 0.063% (23/36,489) (*p* < 0.001) [8]. The low cardiac arrest incidence in this study might suggest several characteristics of the patient population in the unit, including having patients in generally stable condition at the time of arrival or patients requiring lower-risk procedures. The proportion of vascular procedures in our IR unit sample (55.7%) was lower than the proportion reported by a similar study (60.7%) [8], and this difference was statistically significant (*p* = 0.0394).

The patients’ medical state at the time of arrival at the IR unit was assessed, retrospectively, using the ASA classification system. The ASA status distribution of cardiac arrest patients in the unit resembled that of the ACS-NSQIP cardiac arrest patient data, affirming that the data collected in this IR unit is relevant to other studies. The ASA status distribution of cardiac arrest patients in the IR unit was different from that of patients who did not experience cardiac arrest. The median ASA status assigned to cardiac arrest patients was higher than that of non-cardiac arrest patients, and the association between ASA classification and cardiac arrest occurrence was statistically significant. This finding is consistent with prior literature that identified high ASA class as a strong predictor of intraoperative cardiac arrest [3,9,10]. Such distribution is similar to the ASA status distribution among the cardiac arrest patients reported by the ACS-NSQIP (*χ*^2^ = 5.98, df = 3, *p* = 0.11) [7]. Relative risk analysis among the patients with high ASA classifications (III or higher) revealed that a higher ASA class was associated with a greater susceptibility to cardiac arrest. More attention should be paid to the patient’s vital signs, keeping in mind that there may be a higher chance of intraprocedural cardiac arrest among patients with higher ASA classification designations.

The incidence of cardiac arrest was affected by procedure type. A chi-square test of independence revealed a statistically significant association between cardiac arrest occurrence and vascular procedures (*p* = 0.024). Further dividing the procedures into subcategories revealed that the relative risks of non-TACE atrial procedures were 4.4 and 11.7 compared with biliary procedures and percutaneous catheter drainage. Additionally, TACE and other arterial procedures were associated with the highest mortality rates among the cardiac arrest patients in the IR unit. Of the six patients who died before hospital discharge, five (83.3%) underwent arterial procedures. This result was expected because arterial interventions are known to have strong associations with coronary artery disease, which is known to cause 80% of sudden cardiac deaths [8,11,12].

AKI was associated with a significantly lower risk of cardiac arrest relative to malignancy, HTN, and DM. Although an increased risk of cardiac arrest has been reported in association with AKI, malignancy, HTN, and DM [13,14,15,16,17], few studies have been conducted on comparing the varying risk levels of cardiac arrest among patients with such underlying conditions. One plausible explanation to account for the disparity that we observed is that AKI is not a chronic condition [18,19], meaning that, for a patient with AKI, there may have been additional prompt treatments that decreased the relative risk of a cardiac arrest before, and during, the procedure [20,21]. However, additional studies are required to fully understand why AKI patients had a lower risk of cardiac arrest than patients with chronic conditions. Such studies will be beneficial, especially because cardiac arrest events precipitated by non-cardiac causes are often associated with lower rates of survival and recovery than cardiac-origin cardiac arrest cases [22]. The elevated relative risk of cardiac arrest, among patients with DM and HTN in this IR unit, was consistent with the results reported by a similar IR cohort study conducted in Virginia, USA [8], which identified DM (52%) and HTN (43%) as common comorbid illnesses among their cardiac arrest patients.

The survival rate of the patients who experienced cardiac arrest in this IR unit was relatively high. An analysis of the National Registry for Cardiopulmonary Resuscitation (NRCR) data reported that, among 49,130 in-hospital cardiac arrest cases, more than half of the patients (55.2%) failed to survive resuscitation, and the all-cause in-hospital cardiac arrest mortality rate was 82.6% [23]. Unlike the NRCR study, in which most of the patients died in the hospital, the majority (73.9%) of cardiac arrest patients in the IR unit in this study survived until hospital discharge. This proportion was also higher than the cardiac arrest survival rate of 60.7% reported by the IR cohort study in Virginia [8]. The NRCR analysis reported that an additional 1.5% of patients were discharged with central neurologic dysfunction severe enough to categorize them with non-survivors, but none of our resuscitation survivors suffered such adverse brain damage. This disparity in the outcomes was expected, considering that 47.5% and 10.7% of cardiac arrest patients in the NRCR cohort study were from intensive care units and emergency departments, respectively, compared with only 8.7% and 4.3% of the IR unit and cardiac arrest patients in this study.

There were several limitations to this study. This was a retrospective study based on the IR unit’s patient charts. As multiple physicians did the recordkeeping, each case was exposed to the recorder’s recall bias and misclassification, as well as misinterpretation errors committed by the classifier. A representative sample had to be selected because the IR unit lacked a central statistical system. Although the sample size was large enough to minimize the error margin and closely approximate the population parameters, the study has selection bias and inference errors. Another limitation is the low number of cardiac arrest patients included in the study. There were only 23 cardiac arrest cases in the IR unit over the study period, and the scarcity of the event reduced statistical power, which increased the risk of inaccurate extrapolation. Some of the cardiac arrest cases were pediatric cases, which may have added confounding effects that were unidentified in this study. Finally, this study only addressed how each risk factor affects cardiac arrest susceptibility independently. The limited scope of this study leaves several confounding variables unidentified. Studying the confounding nature of the risk factors in future studies will provide further insight.

## 5. Conclusions

Age, procedure urgency, ASA physical status, procedure type, and underlying disease can help predict a patient’s susceptibility to cardiac arrest during an IR procedure.

## Figures and Tables

**Figure 1 jcm-11-00511-f001:**
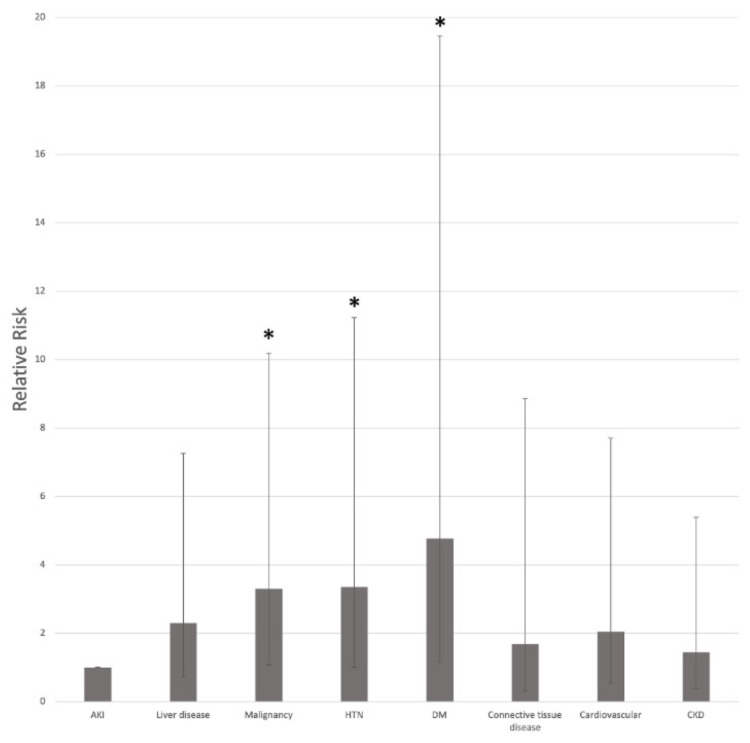
Relative risks of underlying disease, with respect to acute kidney injury. The frequencies of cardiac arrest were calculated for each underlying condition. Relative risk was calculated by comparing the cardiac arrest incidence of a disease against the cardiac arrest incidence of acute kidney injury. The relative risks of malignancy, hypertension, and diabetes mellitus * were significantly higher than that of acute kidney injury. AKI, acute kidney injury; CKD, chronic kidney disease; DM, diabetes mellitus; HTN, hypertension.

**Table 1 jcm-11-00511-t001:** Demographic data of cardiac arrest patients and IR unit sample.

	Cardiac Arrest Patients (N = 23)	IR Unit Sample (N = 400)	Significance
Gender, *n* (%)			*p* = 0.081 *
Male	12 (52.2)	137 (34.3)	
Female	11 (47.8)	263 (65.8)	
Age (years), *n* (%)			*p* < 0.01 ^†^; *χ*^2^ = 24.24; df = 2
Infant (<1 year)	2 (8.7)	1 (0.3)	
Child (1–18 years)	1 (4.3)	4 (1)	
Adult (>18 years)	20 (87)	395 (97.7)	
Mean ± SD	55.61 ± 23.36	62.99 ± 14.21	*p* = 0.133 ^‡^
Procedure schedule			*p* < 0.001 *
Elective	19 (82.6)	394 (98.5)	
Urgent	4 (17.4)	6 (1.5)	
Contrast medium, *n* (%)			*p* = 0.246 *
Used	16 (69.6)	229 (57.3)	
Absent	7 (30.4)	171 (42.8)	
ASA physical status, *n* (%)			*p* < 0.001 ^†^; *χ*^2^ = 104.47; df = 3
I–II	0 (0)	67 (16.8)	
III	13 (56.5)	322 (80.5)	
IV	6 (26.1)	11 (2.8)	
V	4 (17.4)	0 (0)	

ASA, American Society of Anesthesiologists; IR, interventional radiology; SD, standard deviation. * two proportion z-test; ^†^ chi-square test; ^‡^ two sample z-test. Note: The percentages were rounded to the nearest tenth; the values may or may not add up to 100.

**Table 2 jcm-11-00511-t002:** Procedure types among cardiac arrest patients and within the IR unit sample.

	24 Procedures for 23 Cardiac Arrest Patients	404 Procedures for 400 Patients in the IR Unit Sample	Significance
Vascular procedures, *n* (%)	19 (79.2)	225 (55.7)	*p* = 0.013
TACE	5 (20.1)	78 (19.3)	*p* = 0.795
Non-TACE arterial procedure	4 (16.7)	21 (5.2)	*p* = 0.016
Venous procedure	9 (37.5)	121 (30.0)	*p* = 0.368
Arteriovenous fistula	1 (4.2)	5 (1.2)	*p* = 0.222
Non-vascular procedures	5 (20.8)	179 (44.3)	*p* = 0.031
Biliary	3 (12.5)	81 (20.0)	*p* = 0.401
Urologic	0 (0)	18 (4.5)	*p* = 0.298
Percutaneous catheter drainage	1 (4.2)	71 (17.6)	*p* = 0.097
Gastrointestinal	1 (4.2)	9 (2.2)	*p* = 0.522

IR, interventional radiology; TACE, transarterial chemoembolization. Significance was calculated using two proportion z-test. Note: The percentages were rounded to the nearest tenth; the values may or may not add up to 100.

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
