# Peer review of "Cardiac Arrest during Interventional Radiology Procedures: A 7-Year Single-Center Retrospective Study"

_jcm, 2022, doi:10.3390/jcm11030511_

Round 1

Reviewer 1 Report

I believe that recognize nomenclature is cardiac arrest and cardio-pulmonary resuscitation please refer to the guidelines:

Part 1: Executive Summary: 2020 American Heart Association Guidelines for Cardiopulmonary Resuscitation and Emergency Cardiovascular Care

Raina M. Merchant, MD, MSHP, Alexis A. Topjian, MD, MSCE, Ashish R. Panchal, MD, PhD, Adam Cheng, MD, Khalid Aziz, MBBS, MA, MEd(IT), Katherine M. Berg, MD, Eric J. Lavonas, MD, MS, David J. Magid, MD, MPH, On behalf of the Adult Basic and Advanced Life Support, Pediatric Basic and Advanced Life Support, Neonatal Life Support, Resuscitation Education Science, and Systems of Care Writing Groups  

Lott C, TruhláÅ™ A, Alfonzo A, Barelli A, González-Salvado V, Hinkelbein J, Nolan JP, Paal P, Perkins GD, Thies KC, Yeung J, Zideman DA, Soar J; ERC Special Circumstances Writing Group Collaborators. European Resuscitation Council Guidelines 2021: Cardiac arrest in special circumstances. Resuscitation. 2021 Apr;161:152-219. doi: 10.1016/j.resuscitation.2021.02.011. Epub 2021 Mar 24. Erratum in: Resuscitation. 2021 Oct;167:91-92. PMID: 33773826.

Material and Methods:

The description of power analysis is quite unclear. Please provide data for assuming power analysis for this study. Anticipated rate of incidence, Type 1 and type 2 error and power.

What was the method of selection to include patients in the study?

This could be a potentially a very interesting analysis but in the current form it doesn’t bring anything of relevance.

“Age, procedure urgency, ASA physical status, procedure type, and underlying disease can help predict a patient’s susceptibility to CPA during an IR procedure. “

This statement is quite general and could be applied to any cardiac arrest. Do authors have any conclusion and suggestion based on this study that would help prevent cardiac arrest in the IR.

Author Response

Point 1: I believe that recognize nomenclature is cardiac arrest and cardio-pulmonary resuscitation please refer to the guidelines: Part 1: Executive Summary: 2020 American Heart Association Guidelines for Cardiopulmonary Resuscitation and Emergency Cardiovascular Care

Response 1: We adjusted the nomenclature according to your recommendation – ‘Cardiopulmonary arrest’ was changed to ‘Cardiac arrest’.  

Point 2: The description of power analysis is quite unclear. Please provide data for assuming power analysis for this study. Anticipated rate of incidence, Type 1 and type 2 error and power.

Response 2: We thought the P value included for every statistical test would be sufficient to convey the information and significance of the data, and including anticipated rate of incident, type 1 and 2 error and power for all of them would be redundant. We’ve included type 1 and 2 error and power for some of them.

Point 3: What was the method of selection to include patients in the study?

Response 3: The method of selection was a simple random sampling (SRS). This information has been added to material and methods section.

Point 4: This could be a potentially a very interesting analysis but in the current form it doesn’t bring anything of relevance. “Age, procedure urgency, ASA physical status, procedure type, and underlying disease can help predict a patient’s susceptibility to CPA during an IR procedure.“ This statement is quite general and could be applied to any cardiac arrest. Do authors have any conclusion and suggestion based on this study that would help prevent cardiac arrest in the IR.

Response 4: As stated in Discussion and conclusions, it is important to determine whether there are any factors predicting susceptibility to cardiac arrest during an IR procedure

Reviewer 2 Report

The authors presented a retrospective study on the occurrence of in-hospital cardiac arrest (CPA) in the Interventional Radiology department, and it's predictors. I have the following comments:

  1. Table 2 does not contain percentages and p-values (like table 1). Please adjust.
  2. What do the authors mean with "AKI"? What was the definition? Do they mean AKI on beforehand or also as a complication of the IR procedure?
  3. In Figure 1, the confidence intervals are missing.
  4. Did the authors consider a multivariable regression analysis to identify independent predictors of CPA? Why or why not? Think of revising limitations section accordingly.
  5. What are the clinical implications of the study?

Author Response

Point 1: Table 2 does not contain percentages and p-values (like table 1). Please adjust.

Response 1: We adjusted table 2 to include percentages and p-values.

Point 2: What do the authors mean with "AKI"? What was the definition? Do they mean AKI on beforehand or also as a complication of the IR procedure?

Response 2: AKI is acute kidney injury, and it refers to an impairment of the kidney caused by a relatively short, recognizable event, such as a surgery, injury, medication and others. This is different from chronic kidney disease, which is an impairment of kidney caused by long term diseases, such as diabetes and high blood pressure. The patients would have had AKI beforehand prior to the IR procedures, and we noticed in our statistical analysis that patients with AKI have lower risks of having cardiac arrest during an IR procedure than other patients with different underlying disease.

Point 3: In Figure 1, the confidence intervals are missing.

Response 3: We adjusted figure 1 to include confidence intervals.

Point 4: Did the authors consider a multivariable regression analysis to identify independent predictors of CPA? Why or why not? Think of revising limitations section accordingly.

What are the clinical implications of the study?

Response 4: We considered adding a multivariable regression analysis after reading your comment. At the end, we decided not to do a multivariable regression analysis because the relationship between CPA occurrence and multiple independent predictors is not in the scope of what can be analyzed from our cohort study. The limitation section has been adjusted to specify that this study only addressed how each risk factor affects cardiac arrest susceptibility independently.

Point 5: What are the clinical implications of the study?

Response 5: It is important to determine whether there are any factors expected to be associated with cardiac arrest during the IR procedure and reduce these factors as much as possible and prepare for possible cardiac arrest.

Reviewer 3 Report

The authors investigated in this article entitled “cardiac arrest during interventional radiology procedures: a 7-year single-center retrospective study” the risk factors of cardiac arrest during interventional radiology procedures. They found that age, procedure urgency, AS physical status, procedure type, and underlying disease might be able to help predict patients’ susceptibility to cardiac arrest during interventional radiology procedure. Several concerns have been raised.

  1. The abstract section seems to be lacking.

  1. The format of tables are not appropriate. For example, NS, χ2, etc.

  1. Table 1 lacks a variety of important patients’ baseline data.

  1. It might be challenging to analyze combined data including infant/child and adult.

  1. Variable were not adjusted for other potential confounders.

Author Response

Point 1: The abstract section seems to be lacking.

Response 1: The abstract has been updated.

Point 2: The format of tables are not appropriate. For example, NS, χ2, etc

Response 2: All “NS”s are now replaced with p-values,  and the χ symbol has been updated.

Point 3: Table 1 lacks a variety of important patients’ baseline data.

Response 3: Since the study has been done retrospectively based on available information, some of the patients’ baseline data may have been left out. We believe we included essential information that is relevant to the study.

Point 4: It might be challenging to analyze combined data including infant/child and adult.

Response 4: It was challenging to separate adult data from pediatric data since only 23 cases of cardiac arrest occurred during our study period. However, this limitation has been added to the discussion section.

Point 5: Variable were not adjusted for other potential confounders.

Response 5: In this study, we only addressed how each risk factor is associated with cardiac arrest susceptibility independently. We have updated our limitation section accordingly.

Round 2

Reviewer 1 Report

I would suggest that paper would be revised by statistician there are some serious flaws in the study design and methodology.

Just as one example: The average 163 ASA assigned to cardiac arrest patient was 3.61 and that of patient who did not experience 164 cardiac arrest was 2.86 (P < 0.0001).

Why was ASA score used as a continuous variable?

Why in some places there is p-value and in other is simply stated as NS ?

Also, authors did not address issued pointed out by he review.  Most importantly, although authors have analysis interesting and rare events in the IR, there is not enough clinical significant conclusion from this study.   

Author Response

Response to Reviewer 1 Comments

Point 1: Why was ASA score used as a continuous variable?

Response 1: We calculated the average of ASA and conducted a two-sample z test because we treated ASA score as a discrete variable with values ranging from 1 to 5, with 5 being the most severe condition. However, after reading your comment, we recognized that even though the ASA score is a number, the nature of how the numbers were assigned is qualitative and can be seen as categorical rather than a quantitative variable. Therefore, we removed averages and the z-test. Removing this analysis did not change our result or discussion since most of our discussion regarding ASA was done using the Chi-Square test.

Point 2: Why in some places there is p-value and in other is simply stated as NS?

Response 2: We reported all statistically insignificant p-values (P> 0.05) as NS. We meant to report three P-values in table 2 as 0.024, 0.020, 0.024 (instead of 0.24, 0.20, 0.024), but we made a mistake while transferring the data into the chart. The three values have been changed to P< 0.05 in order to keep the format consistent with table 1 when we reported P values as below a certain threshold.

Point 3: Also, authors did not address issued pointed out by he review.  Most importantly, although authors have analysis interesting and rare events in the IR, there is not enough clinical significant conclusion from this study.   

Response 3: It is important to determine whether there are any factors expected to be associated with cardiac arrest during the IR procedure and reduce these factors as much as possible and prepare for possible cardiac arrest.

Reviewer 2 Report

Thank you, sufficiently addressed.

Author Response

Thank you for your review.

Reviewer 3 Report

There are no further comments.